# Musculoskeletal Pain and Non-Classroom Teaching in Times of the COVID-19 Pandemic: Analysis of the Impact on Students from Two Spanish Universities

**DOI:** 10.3390/jcm9124053

**Published:** 2020-12-15

**Authors:** Raquel Leirós-Rodríguez, Óscar Rodríguez-Nogueira, Arrate Pinto-Carral, Mª José Álvarez-Álvarez, Miguel Á. Galán-Martín, Federico Montero-Cuadrado, José Alberto Benítez-Andrades

**Affiliations:** 1Nursing and Physical Therapy Department. Faculty of Health Sciences, University of León, Ave. Astorga 15, 24401 Ponferrada, Spain; rleir@unileon.es; 2SALBIS Research Group, Department of Nursing and Physiotherapy, Universidad de León, 24401 Ponferrada, Spain; arrate.pcarral@unileon.es (A.P.-C.); mj.alvarez@unileon.es (M.J.Á.-Á.); 3Unit for Active Coping Strategies for Pain in Primary Care, East-Valladolid Primary Care Management, Castilla and León Public Health System, 47011 Valladolid, Spain; magalanm@saludcastillayleon.es (M.Á.G.-M.); fmonteroc@saludcastillayleon.es (F.M.-C.); 4SALBIS Research Group, Department of Electric, Systems and Automatics Engineering, Universidad de León, 24071 León, Spain; jbena@unileon.es

**Keywords:** stress, musculoskeletal pain, college students, distance teaching, virtual teaching, COVID-19

## Abstract

The lockdown, due to the coronavirus, has led to a change in lifestyle and physical activity in Spanish university students. The objective of this study was to analyze the prevalence of musculoskeletal pain and changes in physical activity and self-perceived stress in the student bodies of two Spanish Universities during the lockdown. A cross-sectional study was carried out in a sample of 1198 students (70.6% women). The main instruments used for measuring were the Standardized Kuorinka Modified Nordic Questionnaire and the Perceived stress scale (the questionnaire regarding the practice of physical activity). A reduction in the prevalence of musculoskeletal pain (*p* < 0.001) was identified in the sample of men and women, an increase (12.5%) in the frequency of carrying out physical activity from moderate to frequent, and the preference for strength training (15.1%), especially among women, was identified. All of this may be taken into account by health institutions when implementing measures to encourage physical activity in both suitable amounts and types, which improves the quality of life of the students.

## 1. Introduction

The current coronavirus pandemic has given rise to a multitude of profound changes in the daily habits of millions of people. Many of these changes are due to lockdown conditions implemented by countries around the world [1]. Specifically, in Spain, a state of emergency was declared on March 14, 2020, adapting restrictive measures in fundamental rights, such as the free circulation of people, social meetings, the suspension of all public cultural, recreational, and/or sports activities, and the cessation of classroom teaching activities [2]. In regards to the last aspect, the Spanish university community changed its lifestyle habits abruptly and without a period of transition to new teaching models, be it distance and/or virtual [3]. Although each university adapted to the new situation as best as it could, the state’s premises were to maintain the usual schedule, with the previously established workload of hours, with distance, virtual classes, videos, and promoting the use of more autonomous methods of learning by the students. The challenge for the university population went beyond the teachers (who had to modify pedagogical aspects, techniques, and teaching administration) [4,5] to the students (who had to adapt their ways of acquiring skills and abilities in the new scenario) [6,7]. This all took place at the same time as living through an unprecedented situation, which had a huge impact on the emotional health of the students, as indicated in recent research [8,9]. However, no studies were found that determined the impact on the musculoskeletal health of university students during the lockdown period.

At the same time, there is research into the relationship between anxiety, stress, and unsuitable ways of coping with difficult situations and pain [10,11,12]. Musculoskeletal pain in students leads to discomfort itself, which may limit the students’ daily and leisure time, increase psychical stress, and financial costs due to the repetitive use of the specialized health care services. Furthermore, recurrent musculoskeletal pain may impair study performance and affect the students’ future working capacity and health in their transition from university to professional life [13]. Musculoskeletal pain is increasingly common in workers as a result of their exposure to physical, mental, and environmental factors that give rise to the prevalence of musculoskeletal pain [14,15]. Undergraduate students are also exposed to physical and psychological factors, both in the academic setting and in their insertion into the workplace, which could trigger the occurrence of musculoskeletal pain [16].

In their academic routine, students sit for long periods, often in inappropriate chairs, and carry out a large number of curricular activities, which can lead to musculoskeletal overload [17]. Furthermore, they must go to external organizations to attend practical classes in which they carry out professional activities [16]. Students also use laptops and cell phones frequently, both to support their academic activities and in their free time. When using these electronic devices, they tend to adopt inappropriate postures that can cause pain and musculoskeletal alterations, especially in the upper limbs and spine [18]. This is a worrying context, since students are exposed to risk factors, which may increase the occurrence of musculoskeletal pain, and the symptoms can interfere with the well-being and quality of life of these future professionals.

Together with the changes, the lockdown brought with it probable difficulty in implementing physical activity (PA), which is associated with greater rates of musculoskeletal pain [19,20], since in Spain, neither leaving the house nor going out to public spaces was permitted for any activity that was not strictly necessary.

For all of these reasons it was considered necessary to carry out this study with the objectives of: (a) analyzing the prevalence of musculoskeletal pain in students from two Spanish universities prior to and during the period of lockdown; (b) evaluating PA habits and self-perceived stress by the participants during the lockdown; and (c) identifying personal variables (sex, the average hours spent sitting daily, the number of people they lived with at home during the lockdown, and if they used a space to carry out their study activities and virtual classes especially designed for this purpose, or a common area not specified for it), which are related to PA habits during lockdown.

## 2. Experimental Section

### 2.1. Design of the Study and Sample 

A cross-sectional study was carried out on students from two Spanish universities (Universidad de León and Universidad de Valladolid), with a convenience sampling. The total population included in this study was 31,293 students: 18,650 students from the Universidad de Valladolid and 12,643 students from de Universidad de León. It was calculated that, to achieve a 99% degree of confidence and a 4% margin of error, it was necessary to obtain the participation of 1007 students. The criteria for inclusion in the participation were (a) belonging to the student body of the Universities of León or Valladolid; (b) that the said link was, at least, for six months; (c) participating actively in virtual teaching given during the period of lockdown brought about by the COVID-19 pandemic in Spain (that is, between 16th March and 11th May, 2020); (d) to respond to all of the questions included in the evaluation survey and to sign the informed consent for participation in the study in accordance with the Helsinki Declaration (rev. 2013). 

The gathering of data was carried out via email during the months of April and May 2020, by the respective Research Vice-Rectorates, using a digital link, through which the students who wanted to participate voluntarily were able to read and accept the informed consent form, as well as access the questionnaire created for this purpose. To move forward in answering the questionnaire and finalizing it, it was necessary to answer all of the questions included. The data were treated in an anonymous way.

This study was carried out with the approval of the Ethics Committees of the Universidad de León (ETICA-ULE-015-2020) and the Health Area of East Valladolid (PI 20-1787).

### 2.2. Variables in the Study 

The collection of questionnaire data was used as an instrument that included:

(a) The Spanish Standardized Kuorinka Modified Nordic Questionnaire (SNQ): an instrument to analyze the musculoskeletal symptoms in an ergonomic or occupational health context and measure the results of the epidemiological studies on musculoskeletal disorders [21,22], which has a good to very good test–retest reliability (*k* = 0.6–0.81), an internal consistency from good to acceptable (Kuder–Richardson 20 = 0.74–0.87), and a good construct validity in its Spanish version [22]. The SNQ is divided into two parts, the general, and the specific. In this study, only the general part was used, which consists of 27 questions with Yes/No answers about any musculoskeletal symptoms experienced during the previous 12 months or the previous seven days in regards to the impact on activities during the 12 months. All of the questions refer to nine areas: neck, shoulders, elbows, wrists/hands, the upper part of the back, the lower part of the back, hips/thighs, knees, and ankles/feet [21,23]. This part of the questionnaire is modified so that it complies with the circumstances and the context of this research, for this reason, a test was carried out on a convenience sample of 15 people from the University community, who indicated if any aspect of the questionnaire needed to be reworded and if the questions were easy to understand. In this way the questions: “Have you had any problems in the last 12 months (discomfort, unease, or pain) in…?” was modified to “Have you had any problems in the 12 months prior to the lockdown (discomfort, unease, or pain) in…?”. “During the last 12 months have you had any moment in which you were unable to carry out your normal work (at home or outside of it) as a result of the problem?”, was modified to “During the 12 months prior to the lockdown have you had any moment in which you are unable to carry out your normal work (at home or you outside of it) as a result of the problem?”. The question “Have you had a problem during the last 7 days?”, has been modified to “Have you had any problem (discomfort, unease or pain) during the days that we were confined as a result of the coronavirus?” Thus, for the analysis of the results, the total number of painful areas identified was used.

(b) Perceived stress scale (PSS) in its version adapted to Spanish: evaluates the degree to which the people perceive the demands of their environment as unpredictable and uncontrollable; that is, their perception of control over these demands. It had a reliability of (*α* = 0.82, test-retest, *r* = 0.77), and a suitable validity and sensitivity [24,25,26].

(c) Frequency of doing PA: this aspect was polled by means of the questions “Did you do any PA before the lockdown?” and “Did you do any PA during the lockdown?” The options for answering were: (1) no; (2) occasionally (some days a month); and (3) frequently (several days a week).

(d) Type of PA carried out: this aspect was polled by means of the questions “What type of PA did you mainly do before the lockdown?” and “What type of PA did you mainly do during the lockdown?” The options for answering were: (1) none; (2) aerobic; (3) strength exercises; (4) another type of exercise (such as stretching, for example).

(e) Other data of interest: sex, the average number of hours that they spend seated daily, the number of people with whom they lived in their home during the lockdown, and in order to carry out their study activities and participation in virtual teaching, did they use a space especially designed for this purpose or a common area not specified for it.

### 2.3. Statistical Analysis

A descriptive analysis of all the study variables was made through the calculation of the average values (to determine the central tendency) and standard deviation (as a measure of dispersion).

The variables showed a normal distribution according to the Kolmogorov–Smirnov test (p > 0.05), and there was a homogeneity of variances, applying the Levine test. The t-test was used to verify the existence of significant differences between the sexes and in the prevalence of painful areas between the lockdown and the previous year. The ANOVA test with the Bonferroni correction was used to differentiate groups of physical activity divided by sex. A correlation analysis was carried out between PA, the areas that manifested pain and self-perceived stress to find the relationship between them.

We applied the logistic regression model (logit) to analyze the association of independent variables and the dependent dichotomous variable (musculoskeletal pain during the lockdown: 0 = no; 1 = yes). Finally, the adjusted odds ratios (OR) with degree of confidence intervals were estimated using the multivariate regression model. The model was initially adjusted by age. The STATA program, the version 12 statistical package (StataCorp, College Station, TX, USA) was used for the statistical analysis and the statistical meaning was established at a value of *p* < 0.05 for all of the statistical tests.

## 3. Results

Finally, 1198 people participated in the study, of which 846 were women: 70.6% (Table 1). The response rate was 3.8%. Between both sexes, differences were identified in age, in the areas in which pain was perceived during the previous year, the areas of pain limiting working activities during the previous year, the areas of pain perceived during lockdown and stress. These differences indicated that women were significantly younger, and that in all other variables that obtained significantly different results, women had higher results. On the other hand, the number of people with whom they lived, or the use of a specific or common space for academic work, did not come up with different results between the sexes.

The pre- and post-test analysis between the areas of pain during the previous 12 months and the areas of pain during the lockdown obtained significantly lower results during lockdown for the sample as a whole and for the two sample subgroups (*p* < 0.001, for the three tests) (Table 1).

In regards to the analysis of the areas of perceived pain, although they were reduced in the total calculation, in the rachis (cervical dorsal and lumbar regions), the prevalence of musculoskeletal pain increased for the whole of the sample and in both sexes. This increase was significant solely in the prevalence of dorsal pain for the total sample and the subgroup of women (*p* = 0.01, in both cases) (Table 2). On the other hand, the number of participants with pain in the other regions of the body decreased during the lockdown compared to the previous year, except in the subgroup of women who presented pain in both shoulders, which increased (*p* = 0.03).

The frequency of PA carried out changed during the lockdown (contrast pre vs. post-test: *p* < 0.001, for the total sample and the subgroups by sex) (Figure 1). The proportion of women who carried out PA frequently increased by more than 12%. On the other hand, there was an increase of 5.1% in men who had never exercised (*p* = 0.03) because the percentage of men who carried out physical activity, occasionally and frequently, was reduced.

The type of PA taken was also modified during the lockdown (contrast pre- vs. post-test: *p* < 0.001, for the total sample and the subgroups by sex) (Figure 2). The women reduced their practice of aerobic PA (*p* = 0.01) and other activities (*p* < 0.001) and increased the practice of strength training (*p* < 0.001). The men only showed significant changes in the reduction of aerobic exercise (*p* = 0.01) and there was an increase in those who did no exercise at all (*p* = 0.03).

The ANOVA analysis, according to the frequency and type of PA carried out during the lockdown, were not statistically significant for the age, number of hours spent sitting down, and areas of pain during lockdown, nor was stress perceived in the subgroup of men (*p* > 0.05 for all the tests). However, the same ANOVA analysis in the subgroup of women showed different results according to the frequency of carrying out PA by age and the number of people with whom they live. Specifically, between the subgroups who did no PA, and those that did do PA frequently, the former was significantly higher (23.5 ± 7 vs. 21.8 ± 4.2 years old: *p* = 0.01) and they lived with fewer people (2.3 ± 1.1 vs. 2.6 ± 1.2 people: *p* = 0.01). At the same time, the women who carried out no PA also lived with fewer people than those who carried out PA occasionally (2.3 ± 1.1 vs. 2.5 ± 1.2: *p* = 0.03).

The results of the ANOVA analysis were according to the type of PA carried out in the subgroup of women. These obtained statistically significant results for age: the women who carried out no PA at all and those who took aerobic exercise were significantly younger than those who carried out other activities (*p* < 0.001). At the same time, women who carried out no PA were older than those who carried out strength training (23 ± 6.1 vs. 21.7 vs. 3.6 years old: *p* = 0.03).

The number of hours spent sitting down was also significantly different for women who carried out aerobic PA (8.2 ± 2.8 h) in comparison to the women who did not carry out PA (7.3 ± 2.7 h, *p* = 0.04), with those who carried out strength activities (6.9 ± 2.5 h: *p* < 0.001) and those who carried out other activities (6.9 ± 2.6 h: *p* = 0.01).

Finally, women who carried out no PA and those who carried out strength activities during the lockdown differentiated significantly in the number of people with whom they lived (this being fewer in people who did not carry out PA: 2.3 ± 1.1 vs. 2.6 ± 1.1; *p* = 0.01), and in the number of areas in which they referred to pain during the lockdown (this being fewer in people who carried out strength activities: 3.2 ± 2 vs. 3.7 ± 1.9; *p* = 0.01).

The analysis of the correlation of the age variable turned out to be statistically significant from the number of hours spent sitting down (*r* = 0.6; *p* = 0.03) and for the number of painful areas during the lockdown (*r* = 0.68; *p* = 0.01), which is why the older students spent more time sitting down and had more areas of pain. At the same time, the number of hours sitting down was also associated inversely with the frequency of carrying out PA during the lockdown (*r* = −0.8; *p* < 0.01) and directly with self-perceived stress (r = 0.96; *p* < 0.001).

Finally, the frequency of carrying out PA is also directly related with the number of people with whom the individuals lived (*r* = 0.68; *p* = 0.02) and inversely with the number of areas in which they felt pain during the lockdown (*r* = −0.83; *p* = 0.01).

The analysis of the logistical regression indicated that the variables with the capacity to influence the development of musculoskeletal pain were, by order of association: sex (man = 0; woman = 1), age and the frequency of carrying out PA (Table 3). Taking all of these together, there is a direct influence on the probability of developing this type of pain (0.632 < OR > 1.852; *p* < 0.02 all of them). On the other hand, the amount of time spent sitting down daily and the self-perceived stress did not turn out to be significant in the regression model.

## 4. Discussion

The objectives of this research were to analyze the prevalence of musculoskeletal pain in students from two Spanish universities prior to and during lockdown; to evaluate PA habits and self-perceived stress by the participants during the lockdown; and to identify personal variables that are related to their PA habits during the lockdown. In light of the results obtained, musculoskeletal pain reduced slightly among the university students, along with the modification of PA habits, in regards to its frequency and the type of activity carried out. On the other hand, these variables have not been associated, in any case, to the degree of self-perceived stress.

The reduction in the prevalence of musculoskeletal pain, although significant, was very small. This phenomenon is probably related to the age of the study sample (young people), the relatively short period of time that the lockdown lasted, and the low prevalence of musculoskeletal pain that the sample experienced prior to the lockdown. The number of areas of musculoskeletal pain analyzed indicates that women suffered from more areas of pain, both in the year prior to the lockdown and during it, although both gender subgroups stated that it reduced significantly during the lockdown. This greater prevalence of areas of pain is found in other studies, which conclude that sitting down for many hours in front of the device with a screen (as is the case in university students) increases the probability of suffering from musculoskeletal pain significantly more in women than in men [27,28,29]. The differences in incidence and prevalence of musculoskeletal pain between both sexes was identified and highlighted in numerous previous studies [30,31]; however, the evidence available regarding the said differences already existing in the first decades of life are more limited, and are still being researched, although it was already been pointed out that, in adolescents and young adults, lumbar pain is the most frequent, and these symptomatic episodes last longer in women [32,33]. All of this is compatible and in agreement with that identified in this study.

In relation to the increase in the prevalence of pain in the rachis and in the shoulders, previous studies have correlated continuous work with electronic devices, with increasing pain (especially in the aforementioned areas) [34,35,36,37]. It seems that having the keyboard at the height of or below the elbow, and supporting the arms on the tabletop or the armrests of the chair is associated with a lower risk of musculoskeletal problems [38]. Therefore, the change in the type of teaching (from classroom to online), and the increase in time sitting down at home (probably without the ergonomics necessary, regarding the height of the chair position of the armrest) [36,39] may justify the aforementioned findings.

It was identified that the frequency of PA in women increased and, in fact, this could be one of the causes of the decrease in areas of pain during the lockdown, due to the possible effects of carrying out PA on the musculoskeletal systems and the central processing mechanisms of the pain [10,11,12]. In this way, it is probable that the group of women who increased their frequency of carrying out PA during lockdown reached the minimum guidelines of PA recommended for the maintenance of good health [40,41].

The greatest changes in the type of PA came about through the decrease in the practice of aerobic activity in both groups. This phenomenon could be because aerobic activity frequently takes place in the open air (running, cycling, walking, etc.) [42]. Specifically, there was an increase of 15 percentage points in the proportion of women who carried out strength training. This finding, in combination with a significant decrease in the incidence of pain during the lockdown, is congruent with the known benefits to health of vigorous strength training (metabolic, cardiovascular, emotional, and osteomuscular level) [42,43].

On the other hand, the women who carried out no PA during the lockdown were older than those who did PA frequently. This phenomenon could be down to several causes, such as the greater age, the students could most probably have parallel work tasks, which makes free time available to carry out PA more difficult [44,45]. However, the women who did no type of PA at all were older and lived with fewer people (it is probable that, in accordance with that found in other studies, older students prefer to live alone or with fewer people in the latter years of their academic studies, which would justify the few people with whom they live) than those who carried out strength exercises. These data describe some profiles in relation to PA habits: indicating that older women and those that lived alone or with fewer people were the most sedentary, and younger ones and those who lived with more people carried out more PA, more strength training and, at the same time, were those that manifested a lower incidence of musculoskeletal pain. Therefore, the combination of youth, living with several people, and carrying out strength exercises could be key to improving the person and musculoskeletal health during the lockdown. It is necessary to highlight that the behavioral contexts in which the PA is carried out influences its promotion and maintenance [39]. Within this environment, both social–family and emotional [46,47] support have a significant influence, since doing activities in company encourages motivation and the willpower to do exercise [48,49]. This might explain that those students who live with others and carry out exercise would suffer from a kind of motivational “emotional contagion” to achieve a greater habit of carrying out PA together. On the other hand, habit is an important modulator of the maintenance of PA [50], which may explain why those women, who live with other people, increased their frequency of PA.

Finally, the responses highlight the lack of significant results in relation to the levels of stress manifested by the students. This phenomenon could be due to the fact that the relationships existing between the perception of pain and its exacerbation, in accordance with environmental and/or behavioral stimuli, are based on the analysis of patients with intense pain, especially, chronic [51,52]; conditions which, although they have not been collated in the study, are aspects with which the participants probably do not comply.

This study has significant limitations that must be recognized. First, self-reported information on stress and painful areas was used instead of measurements taken by an expert evaluator through objective and validated instruments. Secondly, the information about the frequency and type of PA carried out was also been self-reported rather than using objective information facilitated by accelerometers or pedometers. It would also have been very useful to have an estimation on the stress prior to the lockdown (which has been made through the number of painful areas). Furthermore, the authors acknowledge that the methodology used limits the generalization of the results (invalid outcome measure by changing the SNQ, no causalities can be assessed by cross-sectional studies, no blinding nor recall bias). Those limitations respond to the exceptional situation of lockdown in which the researchers have also found themselves (and of which there was no warning until a few days before its inception). Finally, the population being studied belong exclusively to Spanish Universities, which limits the extension of our results to other populations.

In spite of the aforementioned limitations, there are significant strengths. This is a wide-ranging relational study, which, with caution, is representative of Spanish University students, and truly reflects the situations that they lived through during the lockdown in Spain. At the same time, this is the first time that direct relationships between the lockdown, the PA habits of the University students, and the incidence of musculoskeletal pain have been detected.

## 5. Conclusions

The lockdown lived through between the months of March and May 2020 led to changes in the lifestyle of university students. The increase in frequency of carrying out PA and the preference for strength training, especially among women, has been identified. Simultaneously, a reduction in the prevalence of musculoskeletal pain has been identified.

All of this must be taken into account by health institutions in the face of implementing measures that encourage PA (quantity and in suitable ways) to comply with health regulations, for the promotion of good health and to improve the quality-of-life of the students. All of this highlights the physiological and socio-family differences between both sexes.

## Figures and Tables

**Figure 1 jcm-09-04053-f001:**
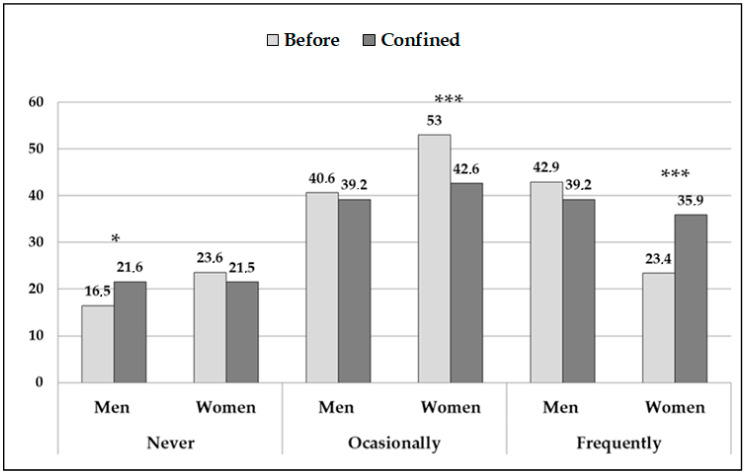
Frequency of physical activity carried out before and after the lockdown. Data provided in %. The asterisks indicate the significant differences in the proportion of men and women within the same response option before and during lockdown: * *p* < 0.05; *** *p* < 0.001.

**Figure 2 jcm-09-04053-f002:**
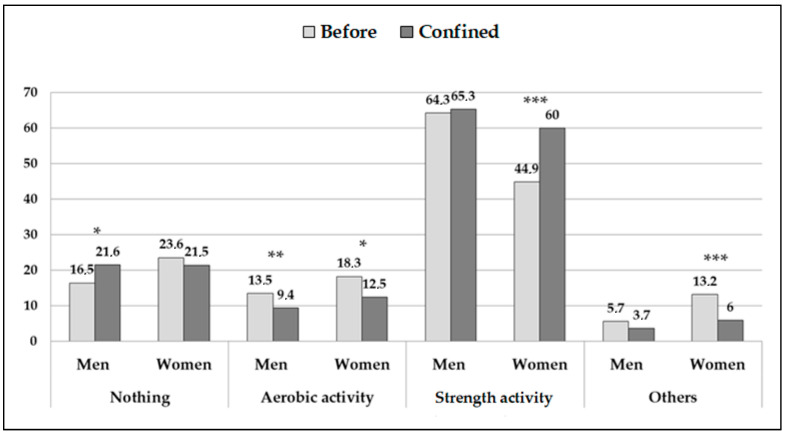
Type of physical activity carried out before and after the lockdown. Data provided in %. The asterisks indicate the significant differences in the proportion of men and women within the same response option before and during lockdown: * *p* < 0.05; ** *p* < 0.01; *** *p* < 0.001.

**Table 1 jcm-09-04053-t001:** Descriptive statistics of the sample (data provided average ± standard deviation).

	All (*n* = 1198)	Men (*n* = 352)	Women (*n* = 846)
Age (years)	22.8 ± 5.9	23.5 ± 7.1 **	22.5 ± 5.3 **
Hours sitting daily (*n*)	7 ± 2.6	6.7 ± 2.7 *	7.1 ± 2.6 *
People with whom lived (*n*)	2.5 ± 1.1	2.4 ± 1	2.5 ± 1.2
PA12 (*n*)	3.7 ± 2.2	3.1 ± 2.2 ***	3.9 ± 2.2 ***
AL12 (*n*)	0.9 ± 1.4	0.6 ± 1 ***	1 ± 1.5 ***
PAC (*n*)	3.2 ± 2	2.7 ± 1.9 ***	3.4 ± 2 ***
PSS (score)	21.9 ± 4.8	20.8 ± 5 ***	22.4 ± 4.6 ***
Workspace (frequency (percentage))			
Specific	885 (73.9%)	254 (72.2%)	631 (74.6%)
Improvised common area	313 (26.1%)	98 (27.8%)	215 (25.4%)

PA12: painful areas during de previous 12 months; AL12: areas with limiting pain during the previous 12 months; PAC: painful areas during lockdown; PSS: Perceived stress scale. The t-test between sex: * *p* < 0.05; ** *p* < 0.01; *** *p* < 0.001.

**Table 2 jcm-09-04053-t002:** Prevalence of pain during the year prior to the lockdown and during it (data provided in (percentage)).

		All (*n* = 1198)	Men (*n* = 352)	Women (*n* = 846)
**Painful Areas During the Previous 12 Months**				
Neck		824 (68.8%)	200 (56.8%)	624 (73.8%)
Dorsal		468 ** (39.1%)	113 (32.1%)	355 ** (42%)
Lumbar		742 (61.9%)	176 (50%)	566 (66.9%)
Shoulders:				
	In one	189 *** (15.8%)	62 (17.6%)	127 *** (15%)
	In both	386 (32.2%)	76 * (21.6%)	310 * (36.6%)
Elbows				
	In one	65 * (5.4%)	20 (5.7%)	45 (5.3%)
	In both	40 * (3.3%)	12 (3.4%)	28 (3.3%)
Hands				
	In one	200 ** (16.7%)	60 (17%)	140 ** (16.5%)
	In both	104 *** (8.7%)	35 *** (9.9%)	69 * (8.2%)
Hips		218 ** (18.2%)	51 (14.5%)	167 *** (19.7%)
Knees		414 ** (34.6%)	110 * (31.3%)	304 * (35.9%)
Ankles		200 *** (16.7%)	63 *** (17.9%)	137 (16.2%)
**Painful Areas During Lockdown**				
Neck		837 (69.9%)	213 (60.5%)	624 (73.8%)
Dorsal		493 ** (41.2%)	122 (34.7%)	371 ** (43.9%)
Lumbar		759 (63.4%)	182 (51.7%)	577 (68.2%)
Shoulders:				
	In one	162 *** (13.5%)	60 (17%)	102 *** (12.1%)
	In both	385 (32.1%)	64 * (18.2%)	321 * (37.9%)
Elbows				
	In one	57 * (4.8%)	18 (5.1%)	39 (4.6%)
	In both	34 * (2.8%)	11 (3.1%)	23 (2.7%)
Hands				
	In one	185 ** (15.4%)	57 (16.2%)	128 ** (15.1%)
	In both	71 *** (5.9%)	17 *** (4.8%)	54 * (6.4%)
Hips		177 ** (14.8%)	49 (13.9%)	128 *** (15.1%)
Knees		377 ** (31.5%)	94 * (26.7%)	283 * (33.5%)
Ankles		159 *** (13.3%)	38 *** (10.8%)	121 (14.3%)

Comparison between painful areas during the previous 12 months vs. during lockdown: * *p* < 0.05; ** *p* < 0.01; *** *p* < 0.0001.

**Table 3 jcm-09-04053-t003:** Binomial logistic regression of musculoskeletal pain in relation to gender, physical activity, and emotional perception.

Variable	OR	SE	*p*	IC 95%
Sex	1.852	0.506	0.02	1.084–3.164
Age	1.141	0.057	0.008	1.035–1.258
Frequency of PA during the lockdown	0.632	0.091	0.001	0.476–0.839
Time spent sitting down daily	1.045	0.048	0.34	0.954–1.143
Self-perceived stress	1.031	0.024	0.19	0.985–1.079

OR: odds ratio; SE: standard error; IC 95%: 95% confidence interval.

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
