# Peer review of "Musculoskeletal Pain and Non-Classroom Teaching in Times of the COVID-19 Pandemic: Analysis of the Impact on Students from Two Spanish Universities"

_jcm, 2020, doi:10.3390/jcm9124053_

Round 1

Reviewer 1 Report

Thanks for inviting me to review this interesting work. This is a needed study but I feel that some changes are necessary, or at least commenting on some important aspects, to make it relevant. See below my main concerns:

Global comments
- I would suggest carefully reviewing all the manuscript writing. For example, between lines 45 and 51 you have two times "However"Introduction
Introduction
- Probably the introduction could benefit from including more research about the incidence of pain in students (types of pain more frequently reported, incidence,...).
- I think PA abbreviation has not been defined.

Design
- How was the survey delivered?
- In analyses section (or measures section), I would explain that a score of number of painful areas was conducted as well as areas with limiting pain during the previous 12 months
Results:
- I would provide the exact direction for these differences and also the tests conducted "The pre-post test analysis between the areas of pain during the 12 previous months and the 148 areas of pain during the confinement obtained significant results for the sample as a whole and for 149 the two sample subgroups (p < 0.001)."- It is not clear who did you compare differences in specific percentages of pre and postconfinement pain in different areas. This should be clearly explained in the analyses section.  The same happens for the differences in percentages of PA.
- I think there is a mistake in the title of Table 2 "Prevalence of errors"- WIth the ANOVAS, is quite confusing with are the dependent and independent variables. As I understand from the objectives, the dependent variable should be the number of pain areas. But there are a lot of ANOVAs performed without a clear objective. I would suggest review this section carefully and rebuild only with meaningful analyses (or change the objective of this study).
- The logistic regression analysis seems more clear. But I wonder what would happen if you would use number of painful areas as the outcome instead of the dichotomous variable created. 
Discussion
- I think that the fact that the decrease of total number of parts was quite small (besides being significant) should be addressed in the discussion.
- In relation to stress, it would have been very useful to have an estimation about the stress prior to the confinement (which has been done by the number of parts). This would have been helped in studying whether there was an increase in stress and if this was related with the increase or not of number of painful areas

Author Response

Dear Editor and Reviewer of the International Journal of Environmental Research & Public Health:

Thank you very much for your suggestions and contributions to improve the quality of the manuscript. Following your indications, we respond, point by point, to the reviewers' comments.

In the text, all the modified or added sentences have been written in red to facilitate the correction by the reviewers.

  1. I would suggest carefully reviewing all the manuscript writing. For example, between lines 45 and 51 you have two times "However".

The authors have reviewed the entire manuscript.

As the response time to resubmit the manuscript is very limited (5 days), if the manuscript is accepted for publication we will forward it to a native translator to thoroughly review the grammar of the text.

  1. Introduction: Probably the introduction could benefit from including more research about the incidence of pain in students (types of pain more frequently reported, incidence,...).

The authors have expanded the introduction with the content you request.

  1. I think PA abbreviation has not been defined.

That abbreviation has been defined.

  1. Design: How was the survey delivered?

The Material and Methods section has been expanded with the information you request.

  1. In analyses section (or measures section), I would explain that a score of number of painful areas was conducted as well as areas with limiting pain during the previous 12 months

The authors have added this detail to the description of the instrument.

  1. Results: I would provide the exact direction for these differences and also the tests conducted "The pre-post test analysis between the areas of pain during the 12 previous months and the 148 areas of pain during the confinement obtained significant results for the sample as a whole and for 149 the two sample subgroups (p < 0.001)."

The description of the Results has been revised and corrected.

  1. It is not clear how did you compare differences in specific percentages of pre and postconfinement pain in different areas. This should be clearly explained in the analyses section.  The same happens for the differences in percentages of PA.
    The authors compared the difference in the number (n) of students who reported pain in each area. The authors did not compare the percentages (it may be that the position of the asterisks in Table 2 next to the percentages invited you to understand otherwise).
    The authors have expanded the description of the Statistical Analysis and modified the position of the asterisks in Table 2.

  1. I think there is a mistake in the title of Table 2 "Prevalence of errors”.

Yes, the authors have corrected the typo.

  1. With the ANOVAS, is quite confusing with are the dependent and independent variables. As I understand from the objectives, the dependent variable should be the number of pain areas. But there are a lot of ANOVAs performed without a clear objective. I would suggest review this section carefully and rebuild only with meaningful analyses (or change the objective of this study).

The authors have added an objective: "(d) to identify personal variables (sex; the average number of hours that they spend seated daily; the number of people with whom they lived in their home during the confinement; and in order to carry out their study activities and participation in virtual teaching, did they use a space especially designed for this purpose or a common area not specified for it) that are related to PA habits during confinement”.

  1. The logistic regression analysis seems more clear. But I wonder what would happen if you would use number of painful areas as the outcome instead of the dichotomous variable created. 

We also tested this possibility and it was not significant in any case.

  1. Discussion: I think that the fact that the decrease of total number of parts was quite small (besides being significant) should be addressed in the discussion.
    The authors have expanded the Discussion by developing this aspect.

  1. In relation to stress, it would have been very useful to have an estimation about the stress prior to the confinement (which has been done by the number of parts). This would have been helped in studying whether there was an increase in stress and if this was related with the increase or not of number of painful áreas.

The authors agree with that assessment. However, we did not find any instrument that evaluated this variable retrospectively.

The authors have added this aspect to the limitations of the research.

Once again, thank you very much for the time spent and the interest shown in this work; as well as in the positive evaluations you have given of it.

Receive a warm greeting,

The authors.

Reviewer 2 Report

A brief summary

The objective of this study was stated by the authors as to analyze the impact of confinement on the prevalence of musculoskeletal pain and its possible relationship with the changes in physical activity and self-perceived stress in the student body of two Spanish Universities. The authors firstly investigated the changes of the prevalence of musculoskeletal pain and changes in physical activity in students during the confinement.

Broad comments

The language has to be corrected throughout the manuscript. The paper addresses an important topic and is of interest. Little is known about the changes of musculoskeletal pain during the confinement. However, the paper needs a lot of cleaning up. There are methodical flaws. The SNQ is explained as a validated tool, but the Spanish version was not validated and in addition changes to the content were made (to adapt to the confinement). The statistical analyses are not concisely explained. Overall, the authors should be more circumspect regarding the findings, discussion and conclusion.

Specific comments to the authors

Abstract

line 20-23:

The study aim is not valid in the provided form. Cross-sectional studies can show a prevalences, but can never prove cause and effect. Therefore, the impact of .... on the prevalence can not be answered by a cross sectional study. Please consider to specify this point.

line 25-28:

Please consider providing the sample size (percentage women), response rate. A mere reduction value with p does not provide an impression of the data (reduced from XX to YY).

line 29-32

The conclusion is not in line with the results of the abstract. Please correct this.

Keywords: the keywords do not match the investigated area. Measured are not: axiety, depression, resilience, or burnout. Please match the keywords to your topic (stress, musculosceletal pain f.e.)

Introduction

Please describe the of virtual teaching of the students (hours per day, interaction, only virtual or mixed teaching etc.)

line 53

Throughout the manuscript: please explain abbreviations when fist used ("PA")

line 56-61

The aim has changes slightly in comparison to the abstract and is less clearly formulated at the end of the introduction. A cross-sectional study can not answer relationships or changes in time. Please be more precise and circumspect.

Experimental Section

This section lacks relevant information, for example: Did the authors perform a sample size calculation, if not: what was the aimed sample size? How were the students approached (recruitment strategy), what was the data collection modus? Was the data collected by paper questionnaires, or online? If online: what software was used? What was the chosen confidence interval? How did they deal with missing data? Was there a adjustment for multiple testing and if not, please be circumspect with the interpretation of "statistically significant" results and the conclusions in the result section and the discussion. This applies for all results.

line 64

There is redundancy: " A cross-sectional, observational study" A cross-sectional study is always an observational study.

Line 75 and following

Please state, if a validated adaptation to Spanish was used (or if at least a native language translation and retranslation was carried out).

The changing of a validated tool (the SNQ, as stated by the authors) invalids a former validation. The changes of the questions are problematic, because it invalids the SNQ general assessment and in addition mixes time periods with perceived causes. For example the changement: From ' “Have you had a problem during the last 7 days?” ' to ' “Have you had any problem (discomfort, unease or pain) during the days that we were confined as a result of the coronavirus?” ' is problematic. The original SNQ asks

line 102-103

The sentence is not intelligible. Please correct the English throughout (f.e. also see line 124-125)

Results

Throughout the results section, "statistically significant" results are reported. Please be circumspect with the language, as this is a cross-sectional study and primary outcome was not specified. There is ongoing discussion in the literature about the use of reporting "statistical significance" in cross-sectional studies and multiple testing.

Line 136: Please provide a response rate (how many students were approache?) For better understanding please provide a flowchart.

Line 137-139

The sentence is not intelligible, please revise the English

This is not a good English: " Except for age, in which the women were significantly younger..."

Line 145:

"PA" stands for 'physical activity' or 'painful areas'? Please explain "PA" as here is explained " PA12: painful areas during de previous 12 months"

line 150

Why did the authors not stick with the chosen p-value (see line 133)? This is confusing. Please make it consistently throughout the manuscript.

Line 155

The sentence is not intelligible.

Table 1: if " statistical meaning was established at a value of p < 0.05 for all of the statistical tests" as stated in line 133-134. Why provide further p-values in line 147?

Line 164

How is it possible in the same population, to reach this increase: "On the other hand, there was an increase of 5% in men who had never taken exercise (p=0.03)." Please explain (was it due to filling out, missing data ...), how this proportion of men could increase.

Discussion

Please consider the comments made earlier, about the language, regarding the English and the more circumspect language regarding the results and conclusion.

Line 292-299

Please include more limitations of the study (multiple testing, invalid outcome measure by changing the SNQ, no causalities can be assessed by cross-sectional studies, no blinding, recall bias etc.) Please discuss, how this limitations might have influenced the results. The sentence is not intelligible, please revise the English throughout: "Firstly, self-reported information has been used the stress has used as regards the areas of pain and stress rather than measurements taken by an expert evaluator."

Author Response

Dear Editor and Reviewer of the International Journal of Environmental Research & Public Health:

Thank you very much for your suggestions and contributions to improve the quality of the manuscript. Following your indications, we respond, point by point, to the reviewers' comments.

In the text, all the modified or added sentences have been written in red to facilitate the correction by the reviewers.

  1. The language has to be corrected throughout the manuscript.

The authors have reviewed the entire manuscript.

As the response time to resubmit the manuscript is very limited (5 days), if the manuscript is accepted for publication we will forward it to a native translator to thoroughly review the grammar of the text.

  1. Abstract. Line 20-23: The study aim is not valid in the provided form. Cross-sectional studies can show a prevalences, but can never prove cause and effect. Therefore, the impact of .... on the prevalence can not be answered by a cross sectional study. Please consider to specify this point.

The authors have reformulated the objectives in the Abstract and in the Introduction.

  1. Line 25-28: Please consider providing the sample size (percentage women), response rate. A mere reduction value with p does not provide an impression of the data (reduced from XX to YY).

The authors have corrected the Abstract and have expanded the description of the study population in Material and Methods and added the response rate at the beginning of Results.

  1. Line 29-32: The conclusion is not in line with the results of the abstract. Please correct this.

The authors have corrected the Abstract.

  1. Keywords: the keywords do not match the investigated area. Measured are not: axiety, depression, resilience, or burnout. Please match the keywords to your topic (stress, musculosceletal pain f.e.)

The authors have modified the Keywords.

  1. Introduction: Please describe the of virtual teaching of the students (hours per day, interaction, only virtual or mixed teaching etc.)

The authors have expanded the Introduction with this information.

  1. Line 53. Throughout the manuscript: please explain abbreviations when fist used ("PA").

That abbreviation has been described.

  1. Line 56-61. The aim has changes slightly in comparison to the abstract and is less clearly formulated at the end of the introduction. A cross-sectional study can not answer relationships or changes in time. Please be more precise and circumspect.

The objectives have been reformulated.

  1. Experimental Section. This section lacks relevant information, for example: Did the authors perform a sample size calculation, if not: what was the aimed sample size? How were the students approached (recruitment strategy), what was the data collection modus? Was the data collected by paper questionnaires, or online? If online: What was the chosen confidence interval? How did they deal with missing data? Was there a adjustment for multiple testing and if not, please be circumspect with the interpretation of "statistically significant" results and the conclusions in the result section and the discussion. This applies for all results.

The Material and Methods section has been expanded to include all the aspects you refer to.

  1. Line 64. There is redundancy: "A cross-sectional, observational study" A cross-sectional study is always an observational study.

That detail has been corrected.

  1. Line 75 and following. Please state, if a validated adaptation to Spanish was used (or if at least a native language translation and retranslation was carried out).

Yes, we use a validated version in Spanish. The authors have corrected the description of the instrument to convey it more clearly.

  1. The changing of a validated tool (the SNQ, as stated by the authors) invalids a former validation. The changes of the questions are problematic, because it invalids the SNQ general assessment and in addition mixes time periods with perceived causes. For example the changement: From ' “Have you had a problem during the last 7 days?” ' to ' “Have you had any problem (discomfort, unease or pain) during the days that we were confined as a result of the coronavirus?” ' is problematic.

The authors recognize the methodological limitation that this implies and is reflected in the limitations of the research (at the end of the Discussion).

At the same time, for the purpose of this research, the authors consider it necessary to adapt certain questions of the SNQ without invalidating all the research carried out.

  1. Line 102-103. The sentence is not intelligible. Please correct the English throughout (f.e. also see line 124-125)

The sentence has been reformulated.

  1. Results. Throughout the results section, "statistically significant" results are reported. Please be circumspect with the language, as this is a cross-sectional study and primary outcome was not specified. There is ongoing discussion in the literature about the use of reporting "statistical significance" in cross-sectional studies and multiple testing.

The description of the results obtained has been thoroughly revised and corrected.

  1. Line 136: Please provide a response rate (how many students were approache?) For better understanding please provide a flowchart.

The response rate has been included at the beginning of Results.

  1. Line 137-139. The sentence is not intelligible, please revise the English

The sentence has been rewritten.

  1. This is not a good English: "Except for age, in which the women were significantly younger...”

The sentence has been rewritten.

  1. Line 145: "PA" stands for 'physical activity' or 'painful areas'? Please explain "PA" as here is explained " PA12: painful areas during de previous 12 months"

PA is physical activity. The abbreviation has been defined the first time it appears in the text.

  1. Line 150. Why did the authors not stick with the chosen p-value (see line 133)? This is confusing. Please make it consistently throughout the manuscript.

To convey the p-values specifically for each analysis performed, the authors specified the p-value in each case. Except when the value of p is lower than 0.001 we use that reference (p <0.001).

There is no inconsistency in that regard throughout the manuscript.

  1. Line 155. The sentence is not intelligible.

The authors have rewritten the sentence.

  1. Table 1: if " statistical meaning was established at a value of p < 0.05 for all of the statistical tests" as stated in line 133-134. Why provide further p-values in line 147?

It is common in scientific articles to transmit the different degrees of significance obtained (as long as they meet the premise of the minimum level of significance established).

It was carried out following the style recommendations of the Journal.

  1. Line 164. How is it possible in the same population, to reach this increase: "On the other hand, there was an increase of 5% in men who had never taken exercise (p=0.03)." Please explain (was it due to filling out, missing data ...), how this proportion of men could increase.

It is possible because the percentage of men who performed physical activity occasionally and frequently was reduced (Figure 1).

This explanation has been added to the text of the manuscript.

  1. Discussion. Please consider the comments made earlier, about the language, regarding the English and the more circumspect language regarding the results and conclusion.

The authors have reviewed the entire manuscript.

As the response time to resubmit the manuscript is very limited (5 days), if the manuscript is accepted for publication we will forward it to a native translator to thoroughly review the grammar of the text.

  1. Line 292-299. Please include more limitations of the study (multiple testing, invalid outcome measure by changing the SNQ, no causalities can be assessed by cross-sectional studies, no blinding, recall bias etc.) Please discuss, how this limitations might have influenced the results.

The authors have expanded the limitations of the research.

  1. The sentence is not intelligible, please revise the English throughout: "Firstly, self-reported information has been used the stress has used as regards the areas of pain and stress rather than measurements taken by an expert evaluator."

The authors have rewritten the sentence.

Once again, thank you very much for the time spent and the interest shown in this work; as well as in the positive evaluations you have given of it.

Receive a warm greeting,

The authors.